# OpenReview forum: "FairGen: controlling fair generations in diffusion models via adaptive latent guidance"
_ICLR.cc/2025/Conference — Submitted to ICLR 2025_

### Official Review · Reviewer_HZfm · 2024-10-30

**Soundness:** 2
**Presentation:** 2
**Contribution:** 3
**Rating:** 3
**Confidence:** 4

**Summary:**

This paper proposes FairGen, addressing the challenge of mitigating generative bias in diffusion models while preserving image quality. It also introduces a bias evaluation benchmark, HBE, covering diverse domains such as employment, education, finance, and healthcare. Empirical results show that FairGen outperforms existing bias mitigation methods, achieving bias reduction while maintaining generation quality.

**Strengths:**

1. FairGen strikes a balance between minimizing bias and preserving image quality.
2. This paper proposes the HBE dataset, which is both comprehensive and well-structured, spanning diverse domains, including underexplored areas.

**Weaknesses:**

1. The writing needs improvement. Paragraph three might be better placed in the Related Work section. The Preliminaries section spends too much space (around 70% of a page) describing the diffusion model, which is not the core content of the paper. Additionally:
- Lines 135 and 223 should use \citep instead of \citet.
- In line 213, prompt_c should be written as prompt c.
- Line 221 contains "mistralai," which should be removed.
- In line 222, a brief description of the HBE dataset should be provided instead of referring directly to Section 4.
- On line 241, the {+1, -1} description is repeated unnecessarily.
2. In Figure 1, the encoder and decoder are not properly described. This is potentially confusing, as the encoder should refer to the text encoder, and the decoder should correspond to the VAE decoder.

**Questions:**

1. Clarification on orthogonality (line 218):
- What exactly does "orthogonality" mean in this context, and why is it used? The sentence feels confusing and could benefit from rephrasing for clarity.
2. HBE dataset split proportions:
- Why does the HBE dataset allocate 40% for training, 10% for validation, and 50% for testing? This allocation seems unusual, as the training set is relatively small while the test set is quite large.
3. Source of the target distribution in Fig. 1:
- What is the source of the target distribution mentioned in Fig. 1? If the correct ratio of attributes (e.g., educated vs. uneducated) is unknown, it seems problematic to assume a 50:50 split. In reality, such a distribution may differ, and it can be difficult to determine the true ratio.

---

> ### Author Response · Authors · 2024-11-22
> **Response to Reviewer Reviewer HZfm**
>
> We thank the reviewer for the feedback and constructive comments. We have carefully considered each point raised and provide detailed responses below.
>
> > Q1: Writing Improvements
>
> We appreciate the feedback on the writing and organization! We revised the text part and also refined the figure in our updated manuscript.
>
>
> Q2 Clarification on Orthogonality (Line 218)
>
> We apologize for the confusion regarding the term "orthogonality." In this context, orthogonality refers to the independence between the guidance for fairness and the original content of the prompt, which also aligns with the prior work [1]. Our method aims to adjust the generation process to mitigate bias without altering the semantic content specified by the user. We want to ensure that the fairness adjustments do not interfere with or distort the user's intended output. Therefore, the guidance vector (which is the difference of two vectors in the original space) would show "orthogonality" to the original vectors.
>
> [1] Wang X, Chen T, Ge Q, et al. Orthogonal subspace learning for language model continual learning[J]. arXiv preprint arXiv:2310.14152, 2023.
>
>
> Q3: HBE Dataset Split Proportions
>
> The allocation of 40% training, 10% validation, and 50% testing in the HBE dataset is designed to prioritize evaluation robustness. Given that our method operates primarily during inference and does not require extensive training data (held for other methods), we allocated a larger portion of the data to testing to thoroughly assess the model's performance across diverse scenarios. This allows us to evaluate the generalization and fairness of the model more comprehensively.
>
>
> Q4: Source of the Target Distribution in Figure 1
>
> The target distribution in Figure 1 is determined based on fairness objectives and societal considerations. In the absence of a known correct ratio for attributes, we assume a uniform distribution (e.g., 50:50 split) as a starting point for fairness. This assumption aligns with principles of equal representation when specific demographic data is unavailable or inapplicable. We acknowledge that in real-world scenarios, the true distribution may differ. Our method is flexible and allows for the target distribution to be adjusted based on reliable data or policy requirements.

---

> > ### Comment · Reviewer_HZfm · 2024-11-26
> >
> > Thank you for your comments and detailed explanations. Most of my concerns have now been addressed.

---

> > > ### Author Response · Authors · 2024-11-27
> > >
> > > Dear Reviewer HZfm,
> > >
> > > Given that most concerns are addressed via rebuttal, we kindly ask if you might reconsider your rating in light of these revisions. We deeply appreciate your time and valuable insights!
> > >
> > > Sincerely,
> > >
> > > Authors of Submission 5559

---

### Official Review · Reviewer_kpaF · 2024-11-03

**Soundness:** 2
**Presentation:** 2
**Contribution:** 1
**Rating:** 3
**Confidence:** 4

**Summary:**

This paper proposes FairGen, a method for controlling the distribution of sensitive attributes, such as gender, in text-to-image generation. FairGen introduces a latent guidance module that calculates the difference between scores conditioned on pairs of prompts differing only in the targeted attributes. A language model is fine-tuned to automatically generate these prompt pairs. Additionally, an indicator guidance module is used to control the guidance direction based on historical statistical properties. A benchmark is also introduced to facilitate fair generation evaluation.

**Strengths:**

The paper tackles fairness in text-to-image generation, a timely and important issue with significant ethical implications. The proposed method is straightforward and shows promising effectiveness. The motivation is clear and well-articulated, and the paper is well-organized and clearly written.

**Weaknesses:**

Some key details are missing, particularly regarding the design of the indicator guidance model $I(c,M,(a\_1,a\_2))$. This component is the core of the proposed method, but it is not adequately described in the main text. Please see some of my question in the Question section.

Another major concern is the inference burden introduced by the method. Since the latent guidance is prompt-dependent, it must be computed at inference time, resulting in three evaluations of the score model per timestep and, therefore, a 300% increase in inference time. Additionally, there are extra inferences required for the language model. For multi-dimensional attributes (as in Eq. (7)), this increase is even more substantial. This high computational cost is problematic, especially when compared to baselines like Shen et al. (2023), which does not increase inference time. Consequently, the comparison may be unfair. To convincingly establish the method's effectiveness, a comparison under equal inference times (e.g., adjusting sampling steps across methods) is needed.

---

Minor:

- L85: Extra space before the period.
- Misuse of \citet and \citep, e.g., L121 and L135.

**Questions:**

- What is the detailed formulation of the indicator guidance model $I(c,M,(a\_1,a\_2))$? Specifically, what statistical properties are used within this model? As clusters are updated over time, how are these statistical properties assigned when KNN is used for re-clustering? Does this imply that all individual historical prompts and their generated properties need to be stored?
- In Eq. (7), each pair of attributes forms a latent guidance. For categorical attributes (e.g., different races), does this mean that a squared number of latent guidances will be incorporated during generation?
- According to the notation $I(c,M,(a\_1,a\_2))\\in\\{1,-1\\}$, it seems that latent guidance is always active. What happens if the prompt already specifies the attribute? For example, with a prompt like "A male computer programmer," will the method still cluster this prompt's features $E(c)$ and include this generation in the statistical counts? If not, how is it determined whether the method applies for a given prompt?
- The text suggests that for each cluster, historical property counts are maintained. Are these counts global statistics, or are they calculated separately for each user? This matters because using global statistics without immediate updates could lead to biased generation in a small window for individual users.
- How are the features of past generations obtained (e.g., determining if each generation is male or female)? Are external classifiers used for this purpose?

---

> ### Author Response · Authors · 2024-11-22
> **Responses to Reviewer kpaF**
>
> We thank the reviewer for the feedback and constructive comments. We have carefully considered each point raised and provide detailed responses below.
>
> > Q1: More Details and More clarifications on the Indicator Guidance Model
>
> We apologize for the lack of clarity regarding the indicator guidance model.
> We expanded Section 3.2 to provide a detailed formulation of the indicator guidance model, to ensure that readers can fully understand and reproduce our method.
> In brief, each cluster represents the set of prompts regarding the same main objective (e.g., computer programmer) and the generation statistics is basically the number of generations for different attributes (males/females/white/nonwhite/....). According to these statistics, we can determine the target attribute for the current generation to achieve a certain target distribution. We use the same attribute detection module as test time (i.e, VQA model). We assume all this information are maintained in a global memory in the central server.
> We also would like to clarify that simialr to existing literature, we do not test such attribute-aware prompts in our benchmark. If we aim to consider such prompt, we can simply have an attribute detection module which enforces no attribute control given a positive detection result.
>
> > Q2: Discussion on Computational Costs
>
> Thank you for pointing out the need to discuss computational costs. We added additional comparisons of runtime in Section 5.4. As a training-free method, FairGen induces no training computational costs. In the inference stage, although FairGen induces $1+k*|\mathcal{A}|$ noises estimates in each diffusion step, where $k$ is the number of bias types we want to control and $|\mathcal{A}|$ is the average number of sensitive attributes, the adaptive guidance is only enforced at a small portion of intermediate diffusion steps (details in Section 5.3.3). Additionally, the noise estimates for different attributes are independent and parallelized in the inference. Therefore, FairGen only leads to marginal runtime overhead compared to the baselines while mitigating the bias significantly.

---

> ### Author Response · Authors · 2024-11-27
> **Official Comments by Authors**
>
> Dear Reviewer kpaF,
>
> As the end of the discussion period is approaching, we would like to gently remind you of our responses to your comments. We wonder whether your concerns have been addressed and appreciate any further questions or comments you might have.
>
> Sincerely,
>
> Authors of Submission 5559

---

### Official Review · Reviewer_15oU · 2024-11-04

**Soundness:** 2
**Presentation:** 3
**Contribution:** 2
**Rating:** 5
**Confidence:** 3

**Summary:**

This paper introduces FairGen, a novel adaptive latent guidance mechanism designed to mitigate biases in text-to-image diffusion models while preserving image quality. FairGen operates during inference, using an adaptive latent guidance module combined with an auxiliary memory module to control the generated output's attribute distribution, allowing for dynamic bias mitigation. The method demonstrates flexibility and efficacy by enabling controlled generation aligned with specific target distributions, making it superior to existing prompt intervention and fine-tuning approaches. To evaluate FairGen, the paper also introduces a comprehensive benchmark, the HBE dataset, which covers diverse domains and complex prompt structures, providing a robust framework to assess generative fairness. Experimental results indicate that FairGen achieves substantial bias reduction while maintaining high image quality, showing promise as an effective solution for fairness-aware generation in diffusion models.

**Strengths:**

1. FairGen’s innovative adaptive latent guidance mechanism allows for dynamic and precise bias control in diffusion models, significantly improving upon existing static or fine-tuning methods.

2. The introduction of the HBE benchmark provides a comprehensive and realistic framework for assessing bias across multiple domains, making it a valuable contribution to future research in generative fairness.

3. Experimental results show that FairGen achieves substantial bias reduction while preserving high image quality, demonstrating its effectiveness in balancing fairness and visual fidelity in text-to-image generation.

**Weaknesses:**

1. The paper does not provide a comprehensive comparison of FairGen with methods beyond the diffusion model domain, limiting insights into its broader applicability in generative fairness control.
2. It lacks detailed error analysis, particularly regarding failure cases or scenarios where FairGen might amplify bias unintentionally.
3. This paper lacks in-depth theoretical analysis.
4. There is minimal discussion on the computational costs associated with integrating the adaptive latent guidance mechanism into large-scale diffusion models.
5. Some intermedium visualization results could benefit the empirical studies.

**Questions:**

see weaknesses

---

> ### Author Response · Authors · 2024-11-22
> **Response to Reviewer 15oU**
>
> We thank the reviewer for the feedback and constructive comments. We have carefully considered each point raised and provide detailed responses below.
>
> > Q1: Comparison to Baselines Beyond Diffusion Models
>
> Thank you for highlighting the need for broader comparisons. While our focus is on text-to-image diffusion models due to their recent prominence and unique challenges, we agree that comparing FairGen with conventional fairness learning could provide valuable insights. Indeed, the finetuning-based method presents one conventional fairness learning algorithm which finetunes the model on a fair data distribution. Via detailed comparisons to it in Section 5.2, we validate that FairGen achieves more fair generations than this strong baseline.
>
>
> > Q2: Lack of Detailed Error Analysis
>
> Thank you for the question! One failure mode of FairGen is that it sometimes induced figures with no target objectives (e.g., humans) and we also find that manipulation of multi-attribute is challenging, which is also analyzed in Section 5.2.
>
>
> > Q3: Discussion on Computational Costs
>
> Thank you for pointing out the need to discuss computational costs. We added additional comparisons of runtime in Section 5.4. As a training-free method, FairGen induces no training computational costs. In the inference stage, although FairGen induces $1+2|\mathcal{A}|$ noises estimates in each diffusion step, where $|\mathcal{A}|$ is the number of sensitive attributes, the adaptive guidance is only enforced at a small portion of intermediate diffusion steps (details in Section 5.3.3). Additionally, the noise estimates for different attributes are independent and parallelized in the inference. Therefore, FairGen only leads to marginal runtime overhead compared to the baselines while mitigating the bias significantly.
>
>
> > Q4: Inclusion of Visualization Results
>
> We agree that intermediate visualizations can enhance the understanding of our method's effectiveness. We add some visualization examples in Section 5.5.

---

> ### Author Response · Authors · 2024-11-27
> **Official Comments by Authors**
>
> Dear Reviewer 15oU,
>
> As the end of the discussion period is approaching, we would like to gently remind you of our responses to your comments. We wonder whether your concerns have been addressed and appreciate any further questions or comments you might have.
>
> Sincerely,
>
> Authors of Submission 5559

---

> > ### Comment · Reviewer_15oU · 2024-11-27
> >
> > Thanks for the feedback. I will keep my score.

---

### Official Review · Reviewer_inXs · 2024-11-05

**Soundness:** 2
**Presentation:** 2
**Contribution:** 2
**Rating:** 5
**Confidence:** 4

**Summary:**

This paper works on reducing bias regarding sensitive attributes such as gender, race, etc., from pretrained Stable Diffusion models. The objective of such a line of work requires reducing bias while preserving the generation alignment with the user prompt. The paper proposes an inference-time guidance strategy (FairGen) that is composed of two components: the adaptive latent guidance module and the indicator guidance module to use the previous generations as the memory to guide the inference process toward producing unbiased images. Specifically, the indicator guidance module is constructed based on the memory of previous generations and the sensitive attributes whose biases are to be removed. The adaptive latent guidance module then takes such output as input, along with other inputs, such as input prompts, to provide guidance for the inference process. For evaluation, the paper proposes a more comprehensive benchmark and showcases superior performance than baselines.

**Strengths:**

- The task that this paper tackles is well-motivated and has attracted the attention of many recent works.
- The proposed inference-time guidance method can be efficient as it does not require modifying the pre-trained Stable Diffusion’s weights and has also achieved effectiveness bias mitigation and quality-preservation results.
- This paper also introduces a comprehensive bias evaluation benchmark HBE that comprises of a wider array of domains, prompt structures, and sensitive attributes than previous benchmarks.

**Weaknesses:**

- **Presentation**. Figure 1 needs to be refined, which is currently misleading as it misses one of the three inputs for the adaptive latent guidance module. From the figure, the module only takes the target attribute and input prompt as inputs, however, it also takes the previous step’s denoised latent result.
- **Effectiveness of DPO**. From the ablation study, the contribution of the DPO component seems minimal when comparing rows 2 and 3 of Table 6.
- **Concern regarding practical significance**. The memory of previous generations that the proposed method is founded on can be unavailable in some practical scenarios where the user only wishes to generate a single or few images. In such a case, since the proposed method is an inference-time strategy that relies on memory, the generations will still be biased.
- **Novelty**. The inference-time guidance strategy for removing biased and unsafe content is already proposed in [1], whose method is similar to the adaptive latent guidance module proposed in this work. Although this paper also uses an indicator guidance module that leverages previous generations as the memory to serve as an indicator for the adaptive latent guidance, such a component is not quite novel as a method.

[1] Safe Latent Diffusion: Mitigating Inappropriate Degeneration in Diffusion Models

**Questions:**

I would really appreciate it if the authors could address the points in the weaknesses section.

---

> ### Author Response · Authors · 2024-11-22
> **Response to Reviewer inXs**
>
> We thank the reviewer for the feedback and constructive comments. We have carefully considered each point raised and provide detailed responses below.
>
> > Q1: Presentation Issue with Figure 1
>
> We appreciate the reviewer highlighting the inconsistency in Figure 1. We acknowledge that the adaptive latent guidance module indeed takes the previous step's denoised latent result as an input, which was not clearly depicted in the original figure. We revised Figure 1 to accurately reflect all inputs to the adaptive latent guidance module, ensuring it includes the target attribute, input prompt, and the denoised latent result from the previous timestep.
>
>
> > Q2: Effectiveness of the DPO Component
>
> Thank you for pointing out the minimal contribution of the DPO component in Table 6. We recognize that the gap gained by DPO appears minimal. However, the DPO component can consolidate the guidance to enhance the awareness of nuanced differences that may lead to effectiveness for target attribute control. We clarified that DPO is not an essential block of FairGen and more sophisticated cases may boost the benefit of DPO training.
>
>
> > Q3: Practical Significance Regarding Memory of Previous Generations
>
> Thank you for the question! According our memory-based indicator guidance module in Section 3.3, we perform conditional generation based on past statistics. In other words, the target attribute is determined such that the generation distribution would be close to the target distribution. Even when the number of generations is small, such mechanism is still applicable. For example, if we aim for balanced gender distribution and the first generation is a male, then we will target a female generation in this case.
>
>
> > Q4: Novelty in Relation to Prior Work
>
> We acknowledge the reference to the work "Safe Latent Diffusion" [1].
> First, we would like to point out that [1] focus on unsafe content moderation, while FairGen aims for unbiased generation. Therefore, [1] cannot be viewed as a direct baseline for comparisons. Second, while both methods employ inference-time guidance strategies, our approach differs significantly in both methodology and application. The adaptive latent guidance module in our work is specifically designed to mitigate biases related to sensitive attributes by dynamically adjusting the latent space during inference. Moreover, our indicator guidance module introduces a novel mechanism that leverages historical generation data to inform the guidance direction, which is not present in [1]. This combination allows for a more targeted and effective bias mitigation strategy.

---

> > ### Comment · Reviewer_inXs · 2024-12-02
> >
> > Thanks for the rebuttal. The response to Q1 has addressed the presentation issue. However, the replies to the other three weaknesses, unfortunately, do not clear my original concerns as they align with my original understanding. For example, in Q2, a component that does not significantly contribute and is not an essential block should be considered for removal for a more streamlined method. In Q3 and Q4, the practical significance and novelty of the mechanism that "leverages historical generation data" is limited. Considering this is the core part of this paper that differs from others (e.g., the overall pipeline is similar to [1] despite acknowledging that this paper also "leverages historical generation data"). In my opinion, this paper is more inclined to reject than to accept. I maintain my original rating of "5: marginally below the acceptance threshold".

---

> ### Author Response · Authors · 2024-11-27
> **Official Comments by Authors**
>
> Dear Reviewer inXs,
>
> As the end of the discussion period is approaching, we would like to gently remind you of our responses to your comments. We wonder whether your concerns have been addressed and appreciate any further questions or comments you might have.
>
> Sincerely,
>
> Authors of Submission 5559

---

### Meta-Review · Area_Chair_qEds · 2024-12-20

**Metareview:**

The paper proposes FairGen, a framework for mitigating biases in diffusion models using an adaptive latent guidance mechanism and a memory-based module. It introduces the HBE benchmark and reports improvements in fairness with minimal impact on image quality. Reviewers found the method's novelty to be limited (e.g. relative to "Safe Latent Diffusion"). Practical applicability also appears to be limited to single-image scenarios by computational overhead compared to baselines.

**Additional Comments On Reviewer Discussion:**

The authors addressed presentation issues and implementation details, but key concerns about novelty and computational efficiency remain unresolved.

---

### Decision · Program_Chairs · 2025-01-22

Reject